# Range-dependent flexibility in the acoustic field of view of echolocating porpoises (*Phocoena phocoena*)

Danuta M Wisniewska[1,2]*, John M Ratcliffe[3,4], Kristian Beedholm[1], Christian B Christensen[1], Mark Johnson[5], Jens C Koblitz[6], Magnus Wahlberg[3,7,8], Peter T Madsen[1]

[1]Zoophysiology, Department of Bioscience, Aarhus University, Aarhus, Denmark; [2]Marine Mammal Research, Department of Bioscience, Aarhus University, Roskilde, Denmark; [3]Sound and Behaviour Group, Institute of Biology, University of Southern Denmark, Odense, Denmark; [4]Department of Biology, University of Toronto Mississauga, Mississauga, Canada; [5]Scottish Oceans Institute, University of St Andrews, St Andrews, Scotland; [6]Animal Physiology, Institute for Neurobiology, University of Tübingen, Tübingen, Germany; [7]Marine Biological Research Centre, University of Southern Denmark, Kerteminde, Denmark; [8]Fjord and Belt, Kerteminde, Denmark

**Abstract** Toothed whales use sonar to detect, locate, and track prey. They adjust emitted sound intensity, auditory sensitivity and click rate to target range, and terminate prey pursuits with high-repetition-rate, low-intensity buzzes. However, their narrow acoustic field of view (FOV) is considered stable throughout target approach, which could facilitate prey escape at close-range. Here, we show that, like some bats, harbour porpoises can broaden their biosonar beam during the terminal phase of attack but, unlike bats, maintain the ability to change beamwidth within this phase. Based on video, MRI, and acoustic-tag recordings, we propose this flexibility is modulated by the melon and implemented to accommodate dynamic spatial relationships with prey and acoustic complexity of surroundings. Despite independent evolution and different means of sound generation and transmission, whales and bats adaptively change their FOV, suggesting that beamwidth flexibility has been an important driver in the evolution of echolocation for prey tracking.

*For correspondence: danuta. wisniewska@bios.au.dk

**Competing interests:** The authors declare that no competing interests exist.

## Introduction

Echolocation has evolved independently multiple times in mammals and birds (*Kellogg et al., 1953*; *Griffin, 1958*; *Konishi and Knudsen, 1979*; *Griffin and Thompson, 1982*), and allows these animals to orient under conditions of poor lighting. However, only toothed whales (henceforth 'whales') and laryngeally echolocating bats (henceforth 'bats') use echolocation to detect, locate, and track prey. In these groups, echolocation signals are primarily ultrasonic (>20 kHz) and are among the most intense biological sounds in water and in air (*Madsen and Surlykke, 2013*). These characteristics mean that in uncluttered spaces these predators can detect small prey many body-lengths ahead of them. Key to increasing the effective sensory range of biosonar is a directional sound beam (i.e., a narrow volume of forwardly ensonified space), which increases intensity along the acoustic axis and reduces ensonification of objects off-axis. It has recently been proposed that the advantages of a narrow sonar beam while in search of prey may have been the primary driver for the evolution of high frequency sonar signals in both whales and bats (*Koblitz et al., 2012*; *Jakobsen et al., 2013*; *Madsen and Surlykke, 2013*).

**eLife digest** Bats and toothed whales such as porpoises have independently evolved the same solution for hunting prey when it is hard to see. Bats hunt in the dark with little light to allow them to see the insects they chase. Porpoises hunt in murky water where different ocean environments can quickly obscure fish from view. So, both bats and porpoises evolved to emit a beam of sound and then track their prey based on the echoes of that sound bouncing off the prey and other objects. This process is called echolocation.

A narrow beam of sound can help a porpoise or bat track distant prey. But as either animal closes in on its prey such a narrow sound beam can be a disadvantage because prey can easily escape to one side. Scientists recently found that bats can widen their sound beam as they close in on prey by changing the frequency—or pitch—of the signal they emit or by adjusting how they open their mouth.

Porpoises, by contrast, create their echolocation clicks by forcing air through a structure in their blowhole called the phonic lips. The sound is transmitted through a fatty structure on the front of their head known as the melon, which gives these animals their characteristic round-headed look, before being transmitted into the sea. Porpoises would also likely benefit from widening their echolocation beam as they approach prey, but it was not clear if and how they could do this.

Wisniewska et al. used 48 tightly spaced underwater microphones to record the clicks emitted by three captive porpoises as they approached a target or a fish. This revealed that in the last stage of their approach, the porpoises could triple the area their sound beam covered, giving them a 'wide angle view' as they closed in. This widening of the sound beam occurred during a very rapid series of echolocation signals called a buzz, which porpoises and bats perform at the end of a pursuit. Unlike bats, porpoises are able to continue to change the width of their sound beam throughout the buzz.

Wisniewska et al. also present a video that shows that the shape of the porpoise's melon changes rapidly during a buzz, which may explain the widening beam. Furthermore, images obtained using a technique called magnetic resonance imaging (MRI) revealed that a porpoise has a network of facial muscles that are capable of producing these beam-widening melon distortions.

As both bats and porpoises have evolved the capability to adjust the width of their sound beam, this ability is likely to be crucial for hunting effectively using echolocation.

As whales and bats close in on prey, both are thought to concurrently decrease signal intensity and auditory sensitivity to partially compensate for reduced transmission loss (*Hartley, 1992*; *Au and Benoit-Bird, 2003*; *Supin et al., 2004*; *Linnenschmidt et al., 2012*; *Madsen and Surlykke, 2013*), while increasing the signal emission rate for faster updates on prey location (*Griffin, 1958*; *Morozov et al., 1972*; *Au, 1993*). Both groups terminate prey pursuits with a low intensity, high repetition rate sequence of echolocation signals called a buzz (see ref. [*Madsen and Surlykke, 2013*] for review). Signal production rates characteristic of buzzes have likely evolved in these echolocators to facilitate close range prey tracking. However, for a given sonar beam, effective beam diameter will decrease as the distance between predator and prey diminishes. Thus, while a directional sound beam enables longer detection range by restricting the acoustic field of view (FOV), it may be disadvantageous to the echolocator at short ranges, as moving prey may easily vanish at the periphery of the FOV. Directionality increases with aperture size (e.g., a bat's gape) and signal frequency (*Au, 1993*) and both factors appear to be exploited by some bats to modify their beam (*Surlykke et al., 2009*; *Jakobsen and Surlykke, 2010*).

Unlike bats, whales do not generate echolocation signals in the larynx nor do they emit them through the mouth or the nares (*Ridgway et al., 1980*; *Cranford et al., 1996*). Instead, whales have evolved specialized sound-producing structures, the phonic lips, located high in the blowhole (*Figure 1*). An echolocation click is generated by pneumatic actuation of the phonic lips as pressurized air is forced past them (*Cranford et al., 1996*). The resulting sound pulse propagates into the fatty melon and is transmitted into the water as a directional beam (*Au et al., 1986*). While the exact function of the melon is not known, its size and properties are expected to affect the radiation pattern of sound from the head (*Varanasi et al., 1975*; *Aroyan et al., 1992*; *Harper et al., 2008*). As the bulbous melon fills a large proportion of the forehead (*Figure 1*), the diameter of the head has come to be considered indicative of radiating aperture size (*Au et al., 1999*).

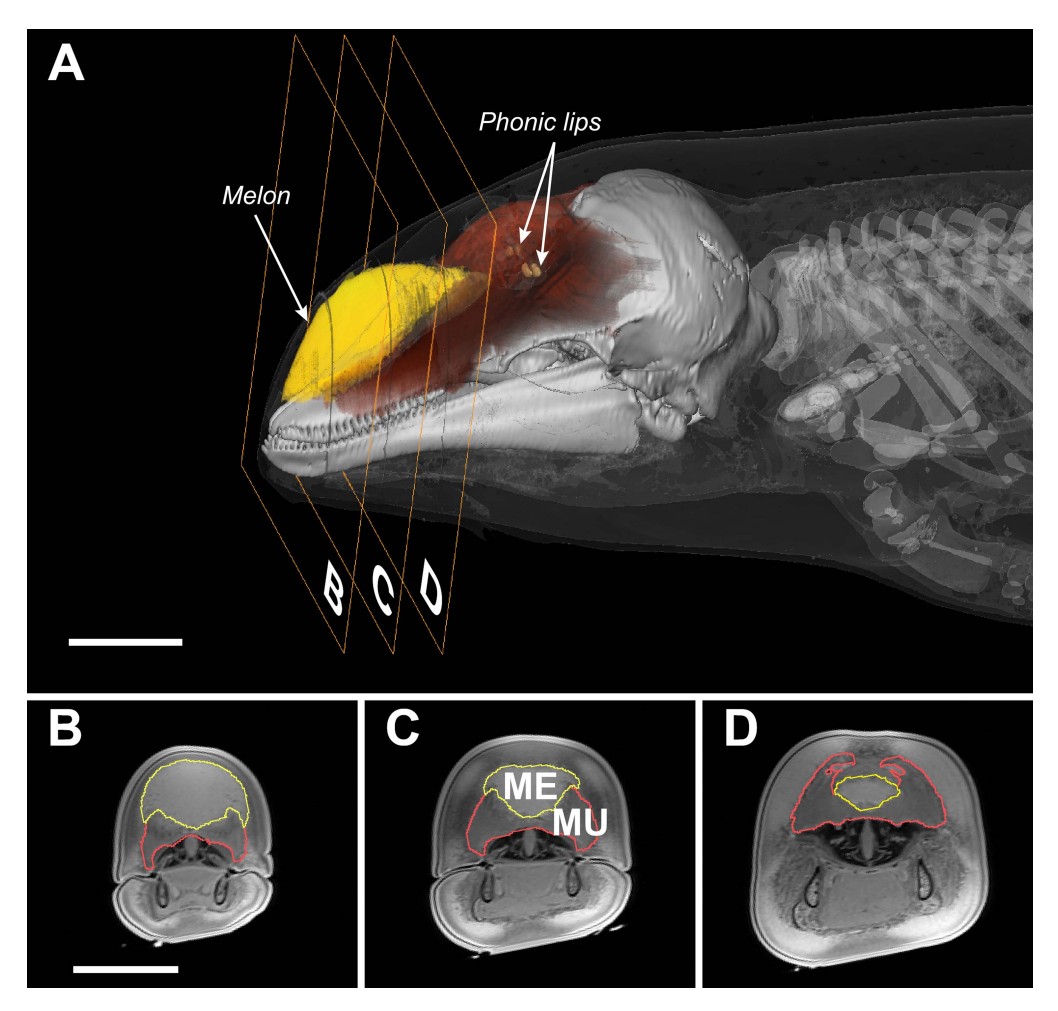

**Figure 1**. Transverse MRI scans of a young harbour porpoise. The scale bars indicate 5 cm (bar in **B** applies also to **C** and **D**). The caudal part of the melon (yellow, **A**) abuts against layers of connective tissue, muscles, and tendons (red, **A**) forming a dense theca, which, along with the skull and a collection of nasal air sacs, reflect the vibrations that originate in the phonic lips (light brown, **A**) into the melon (*Aroyan et al., 1992*; *Cranford et al., 1996*). The melon is under control of highly developed facial musculature (*Harper et al., 2008*; *Huggenberger et al., 2009*). The fibers and tendons of the muscles (Mu) associated with the melon (Me) lie at oblique angles relative to the frontal, transverse, and sagittal body planes (*Harper et al., 2008*; *Huggenberger et al., 2009*). The actions of these richly innervated muscles can change the three-dimensional shape and/or stiffness of the melon (*Harper et al., 2008*; *Huggenberger et al., 2009*), and thus likely adjust the properties of the emitted sound.

Compared to bats, whales exhibit little flexibility in the frequency content of their echolocation signals (*Madsen and Surlykke, 2013*). We might therefore assume whales are capable of only small changes in beam directionality (*Au et al., 1995*) and thus be limited to a rather fixed FOV (*Au et al., 1999*). Yet, given that whales echolocate prey over much longer ranges than bats (*Madsen and Surlykke, 2013*) and are capable of making much narrower beams (*Au et al., 1999*; *Surlykke et al., 2009*; *Koblitz et al., 2012*; *Jakobsen et al., 2013*), such a constraint is hard to reconcile with the disadvantages associated with a fixed beamwidth during close tracking of prey. In bats, pursuit is often over quickly, often with less than 500 ms elapsing between prey detection and interception (*Kalko, 1995*). Prey pursuits by whales can last many seconds (*Johnson et al., 2004*) (*Figure 2*). Accordingly, we hypothesize that long buzzes in which the whale might follow its target from an uncluttered water column to a highly cluttered sea floor (*Figure 2*), and back again, may demand greater beam plasticity than is found in bats.

Corroborating this hypothesis, most of the reported estimates of toothed whale beam patterns show relatively large variability (*Evans, 1973*; *Au et al., 1986*, *1987*, *1995*; *Koblitz et al., 2012*). Furthermore, a bottlenose dolphin was recently observed to steer, and modify the width of, its sonar beam when stationed on a bite plate and presented with targets displaced by large angles with respect to its body axis (*Moore et al., 2008*). The dolphin was proposed to use two mechanisms as means of beamwidth modulation: (i) phase shifting between two pairs of phonic lips dually actuated for generation of a single click and/or (ii) manipulation of the volume and geometry of the melon and the associated air-sacs. However, the latest experimental data suggest that dolphins use a single pair of phonic lips to produce echolocation signals (*Au et al., 2012*; *Madsen et al., 2013*; *Finneran et al., 2014*) and that the strong amplitude dependence of dolphin click spectra gives rise to a variable beam pattern, with more directional signals at higher source levels (*Finneran et al., 2014*). Hence, it remains unclear how and under what circumstances a bottlenose dolphin modulates the width of its sound beam, and whether other whale species, with more stereotyped sonar signals, such as porpoises, are capable of similar beamwidth changes.

In this study, we set out to test the hypothesis that whales can change their FOV adaptively during prey interception. To that end, we used two complementary experiments to record the beam pattern and signal frequency content of echolocation clicks from harbour porpoises (*Phocoena phocoena*) closing in on targets. Additionally, we obtained magnetic resonance imaging on a dead harbour

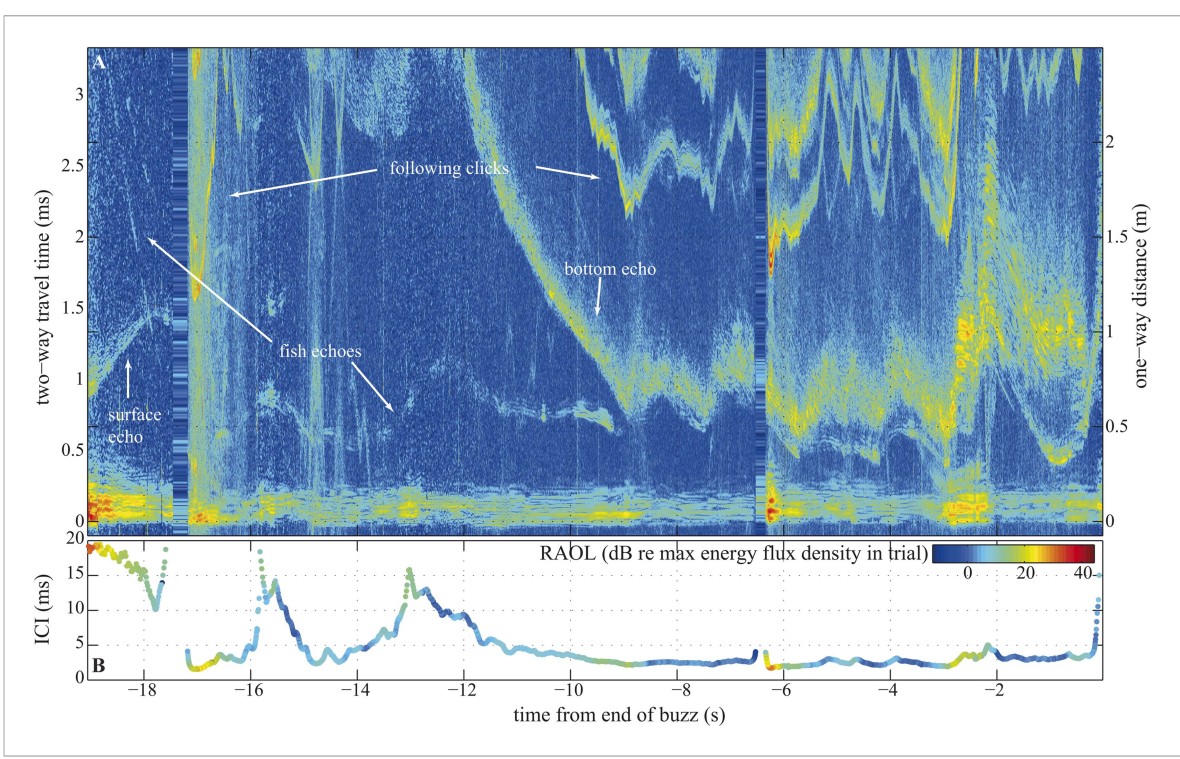

**Figure 2**. Long terminal phase of prey pursuit by an echolocating harbour porpoise. (**A**) Echogram (see 'Material and methods': Live prey capture) displaying sonar clicks and echoes recorded by an acoustic tag attached to the animal just behind its blowhole (*Johnson et al., 2004*). y-axis (left) indicates time elapsed from emitted clicks to returning echoes, expressed also as target range (right). Clicks emitted at rates corresponding to inter-click intervals shorter than 3.3 ms time-window are displayed repeatedly. The color scale indicates signal energy from blue (faint) to red (intense). As pursuit proceeds from water column (0.7–1.1 m from surface, 2–2.5 m above sea floor) to sea bottom the immediate acoustic scene becomes more cluttered with complex bottom echoes shortly following fish echoes (from −9 s onwards). (**B**) Inter-click intervals color-coded for relative apparent output level (RAOL; [*Wisniewska et al., 2012*]) of signals as recorded by the tag. RAOL variation may stem from rapid head movements, source level adjustments, and/or beam directionality changes (with less energy reaching the tag from a narrow beam). On two occasions, when the fish escaped into the open space of the water column (at −16 s and −13 s), the porpoise increased its ICIs significantly, beyond the values considered as buzz (*Wisniewska et al., 2012*). However, when the fish escaped to similar distances while being at the bottom (and thus moving in arguably a more predictable way) the porpoise increased the ICIs only slightly, which might point to anticipatory acoustic tracking on the part of the echolocating animal.

porpoise to visualize the sound generating structures, the fatty melon and the associated musculature and present video demonstrating melon deformations in an actively echolocating harbour porpoise.

## Results

In experiment 1, three captive porpoises were recorded individually using a linear horizontal array of 8 hydrophones spaced 60 cm apart and submerged 75 cm below the water's surface, as the porpoises captured dead fish (*Video 1*). This setup (*Figure 3A*) allowed us to quantify the animals' beam patterns over long ranges as they approached and intercepted natural targets free-floating in the water column. A total of 16 trials were recorded for the three animals (5–6 trials/porpoise). Only trials in which the porpoises swam directly towards the centre of the array were analyzed, amounting to two trials for Freja and Eigil and one trial for Sif (a total of 75 clicks). Data from the two porpoises, Freja and Eigil, recorded during buzzes show that they could up to triple the width of their sonar beam as they approached prey (median difference of 7.85° for four trials: from a mean of 12.9° (6.6°–30.5°, with the large beamwidths produced at short target ranges, *Figure 3B*) during regular clicking (ICI >13 ms; [*Wisniewska et al., 2012*]) to 19.3° (10.2°–28.9°) during buzzes (ICI ≤13 ms; [*Wisniewska et al., 2012*]), *Figure 3B*). However, limitations in the linear array recordings prevented us from drawing strong conclusions about the exact extent of beam widening: the angular resolution was 4–12° and there was no way to precisely determine the vertical direction of the porpoise beam. Even though the video observations indicated that the vertical direction of the porpoise head was not changing with range to the array, this could not be adequately quantified.

We therefore designed experiment two to measure the beamwidth at close target ranges using a non-uniform 2D hydrophone array. One of the three porpoises, Freja, was trained to approach a stationary target surrounded by a star-shaped array of 48 hydrophones in two configurations (*Video 2*). Initially, the hydrophones were arranged at increasing intervals away from the array center to cover the full extent of the beam at relatively long ranges while providing more resolution around the target at short ranges (*Figures 3C, 4*, *Figure 4—figure supplement 1A*). We then repeated the experiment using an array of more-tightly spaced hydrophones (*Figures 3C, 4*, *Figure 4—figure supplement 1B*) and suspended the target further in front of the array, to increase signal intensity resolution at short ranges.

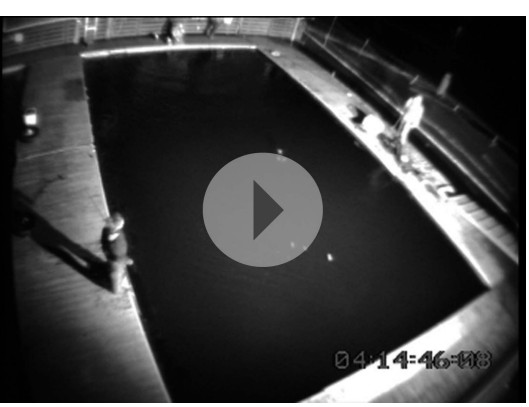

**Video 1.** A representative trial from experiment one. Video shows a porpoise capturing fish in front of the linear hydrophone array. Hydrophones were lowered to a depth of 75 cm along the short side of the pool. Prior to the experiment the blindfolded (i.e., wearing opaque silicone eyecups) porpoise was stationed at the opposite end of the pool. As freshly thawed fish were introduced approx. 3 m from the array, the animal was cued to perform the capture task. The porpoise did not roll throughout the approach. Thus, the observed beam changes could not result from the beam not being rotationally symmetric and the animal rolling consistently during the buzz.

We recorded 123 trials with the long-range configuration and 93 trials with the short-range setup. Only trials in which the porpoise swam directly toward the target (within ±15° vertically and horizontally from the array's center) and repeatedly scanned its beam over the center hydrophone were analyzed. This resulted in 11 trials for the long-range- and 4 trials for the short-range array configuration. For each recorded click, the energy levels received at the hydrophones were fitted to the beam pattern of a circular piston (*Au et al., 1987*; *Møhl et al., 2003*). Only beamwidth estimates based on fits with $R^2 > 0.8$ (see *Figure 4—figure supplement 1*) were considered in the analysis, rendering a total of 34 and 458 clicks for the long-range- and short-range array, respectively. The results of this experiment show that the porpoise could almost double its beamwidth in degrees when switching to a buzz (ICI ≤13 ms; [*Wisniewska et al., 2012*]) at short target ranges (*Figure 3D*). That is, from a mean half-power beamwidth (i.e., the off-axis angle at which the sound energy has decreased by 3 dB relative to the on-axis energy) of 9.1° (6.7–12.3) to a maximum of 15.1° (mean = 11.0, min = 8.2). In this way the animal increased the ensonified area

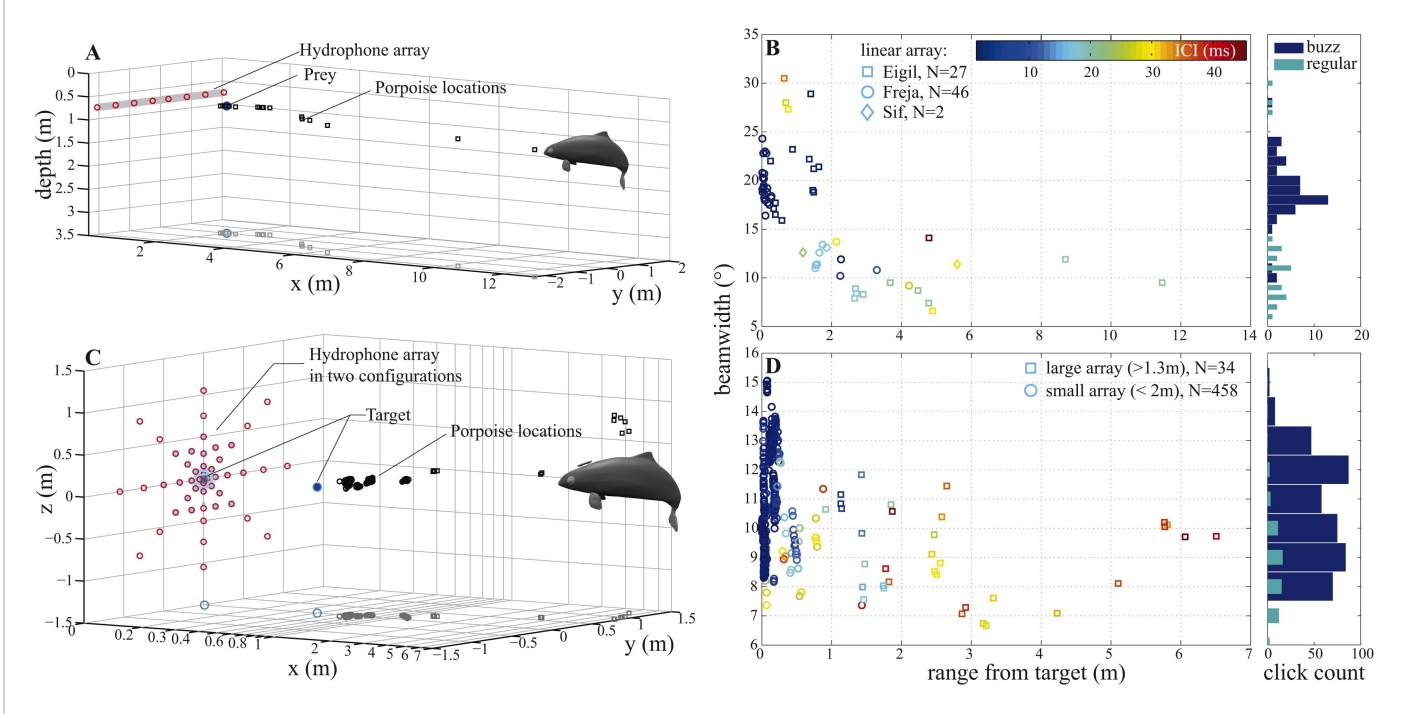

**Figure 3**. Porpoise biosonar beam widens at short target ranges. 3 dB beamwidth recorded in two experimental setups: (**A**, **B**) three harbour porpoises closing on prey and (**C**, **D**) a harbour porpoise approaching an aluminum sphere target. (**A**, **C**) show reconstructed porpoise locations for clicks fulfilling inclusion criteria in one trial per array configuration. Targets and their projections in the x–y plane are marked with dark-blue filled and open circles, respectively. For the small array recordings (light blue in **C**), the target was displaced outward to 0.4 m from the array to maintain high spatial resolution at short ranges. (**B**) Data collected using a horizontal array with effective angular resolution (EAR) of ~12° at ranges of target interception (N = 75). Data points from Freja, the porpoise participating in experiment two, are represented with circles. (**D**) Data gathered with star-shaped arrays in two configurations: large (red in **C**), for long-range recordings (>1.3 m from array to sound source, squares, N = 34) and small (light blue in **C**), for greater resolution at short ranges (<2 m from array to sound source, circles, N = 458) (see **Figure 4—figure supplement 1** for a detailed view of hydrophone spacing in the two array configurations). Hydrophone spacing provided EAR of ~5° at the shortest ranges from the source examined. Color in (**B**) and (**D**) indicates inter-click intervals (ICI), with buzz starting at 13 ms (**Wisniewska et al., 2012**). Buzz- and regular-click datasets in (**D**), used at short- and long-ranges, respectively, have different distributions, but similar medians, because during buzzes the animal repeatedly changed its beamwidth (**Figure 6**). Beam of the long-range clicks varied less and is better approximated by the median.

ahead of it by up to three times (200%) in the terminal buzz phase of its approach to a target, as indicated by the ratio of the area measured at a given target range to the area predicted for that range based on the median −3 dB beam angle calculated for long ranges (i.e., predicted area had the porpoise not switched to 'wide-angle view', **Figure 5**). A significant inverse relationship between the beamwidth of the clicks and distance they were emitted from the target was found for data sets from each experiment (Experiment 1: $F = 56.9$, $R^2 = 0.44$, $p < 0.001$; Experiment 2: $F = 23.6$, $R^2 = 0.05$, $p < 0.001$). Data from experiment two were also analyzed using ANOVA (see 'Materials and methods'). The three clusters (sorted using beamwidth values) differed significantly with respect to distance to target ($F = 5.21$, $p < 0.006$). Specifically, clicks in the group with the greatest beamwidths were emitted significantly closer to the target than clicks in the group with the narrowest beamwidths (Tukey-Karmer HSD, $p < 0.01$).

The timeline of these changes (**Figure 6A,C**) shows that the porpoise varied the width of its beam within the buzz independently of the inter-click intervals (ICIs), with both narrow and wide beam angles used at the lowest ICIs (**Figure 6A,D**). Consequently, the short- and long-range datasets have different distributions, but similar means, and the long-range clicks are better described by the mean beamwidth value (**Figure 3D**). The click centroid frequency (fc) dropped by about 1% at the start of a buzz (from a long-range median of 130.4 kHz (127–136.1) to a median of 127.7 kHz (124.1–134.3), see the color scale in **Figure 6** and **Figure 6—figure supplement 1**).

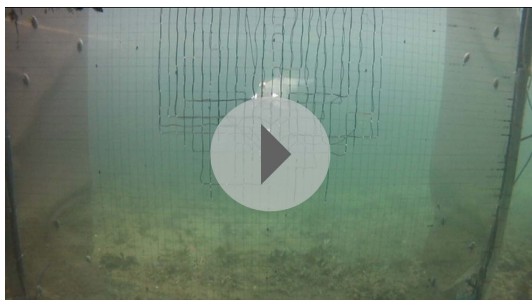

**Video 2.** A representative trial from experiment two. Video shows a blindfolded porpoise closing on an aluminum target in front of a 48-element hydrophone array. The sequence was recorded in a short-range trial, that is, with the array extending to 0.5 m on either side of the centre hydrophone and the target moved to 0.4 m from array centre. Only clicks recorded when the porpoise was <2 m away from the array, at an angle of <15° to its centre and with acoustic axis within 6 cm from the centre hydrophone were selected for the beam-width analysis. The video's soundtrack was replaced with audio recording from the camera-synchronized DTAG-3 carried by the porpoise.

This negligible drop in frequency cannot explain the large changes in beamwidth (*Figure 6—figure supplement 1*; Spearman's correlation tests between centroid frequency and beamwidth: Experiment 1: p = 0.76, Experiment 2: p = 0.67). Rather, the animal must vary the size of the effective aperture, likely by effecting rapid muscular deformations of the melon (*Video 3*), the position of the phonic lips and/or the size and position of the associated air sacs. To visualize the musculature surrounding the melon, we obtained magnetic resonance imaging (MRI) on a dead juvenile harbour porpoise. The scanning images reveal nasal structures that include a complex, richly innervated (*Huggenberger et al., 2009*) network of facial muscles (*Figure 1*). These muscles are homologous to the muscles dedicated to facial expressions in primates, and should enable fast and subtle changes in the shape of nasal components and their associated air sacs (*Huggenberger et al., 2009*).

## Discussion

Here, we show that harbour porpoises can broaden their biosonar beams in the final phase of target approach (*Figure 3*), and, unlike echolocating bats, that they are also able to change their beamwidth within the terminal buzz (*Figure 6*). At its broadest, the beamwidth ranged from 15° in experiment two to more than 30° in experiment one. A number of factors may have contributed to this variability including: (i) differences in effective angular resolution (EAR) between the eight-hydrophone linear array and the 48-hydrophone star-shaped array, (ii) differences in the animals' behaviour when approaching a slowly sinking fish vs a stationary aluminum sphere, and, finally, (iii) potentially higher clutter originating from the large array behind the target and motivating the porpoise to use a narrower beam. Porpoises can adjust their buzz clicking rate to prey range differently when following a fish in open water than when tracking one in the cluttered and more restricted space close to the sea floor (*Figure 2*). The porpoise behaviour and experimental context seem the most likely explanation for the observed differences between the two experiments.

This is the first demonstration, to our knowledge, of a whale controlling its acoustic FOV while actively approaching a target. This control is achieved independently of spectral adjustments (*Figure 6* and *Figure 6—figure supplement 1*), probably by changing the conformation of the melon (*Figure 1*; [*Huggenberger et al., 2009*]), the position of the phonic lips (*Cranford et al., 2014*) and the size and shape of the associated air sacs (*Aroyan et al., 1992*). Due to its heterogeneous structure (*Norris and Harvey, 1974*; *Varanasi et al., 1975*), the melon has long been considered an acoustic impedance matcher that minimizes the reflections and energy loss at the tissue–water interface (*Norris and Harvey, 1974*) and that provides directionality in the emitted click (*Au et al., 1999*, *2006*). Porpoises have a complex facial musculature (*Figure 1*) with nervous innervation with 4.5 times more neurons than human facial muscles (*Jacobs and Jensen, 1964*), leading to the recent proposition that muscle induced deformations of the nasal soft structures such as the melon may provide means to change the FOV (*Huggenberger et al., 2009*). Our acoustic measurements and observations support that hypothesis by demonstrating that the melon and accessory structures apparently operate as the functional equivalent of an adjustable collimating lens of a flashlight. In other words, the porpoise's beam can be dynamically changed from spotlight to floodlight (and everything in between) to best suit the circumstances, offering unprecedented flexibility in control of the FOV in an echolocating animal that is unmatched in visual mammals (*Land and Nilsson, 2012*).

The porpoise's ability to change its FOV within a buzz (*Figure 6*) implies an even greater flexibility than recently documented in vespertilionid bats (*Surlykke et al., 2009*; *Jakobsen and Surlykke, 2010*; *Jakobsen et al., 2013*). While bats adjust their beams to the environment in which they are

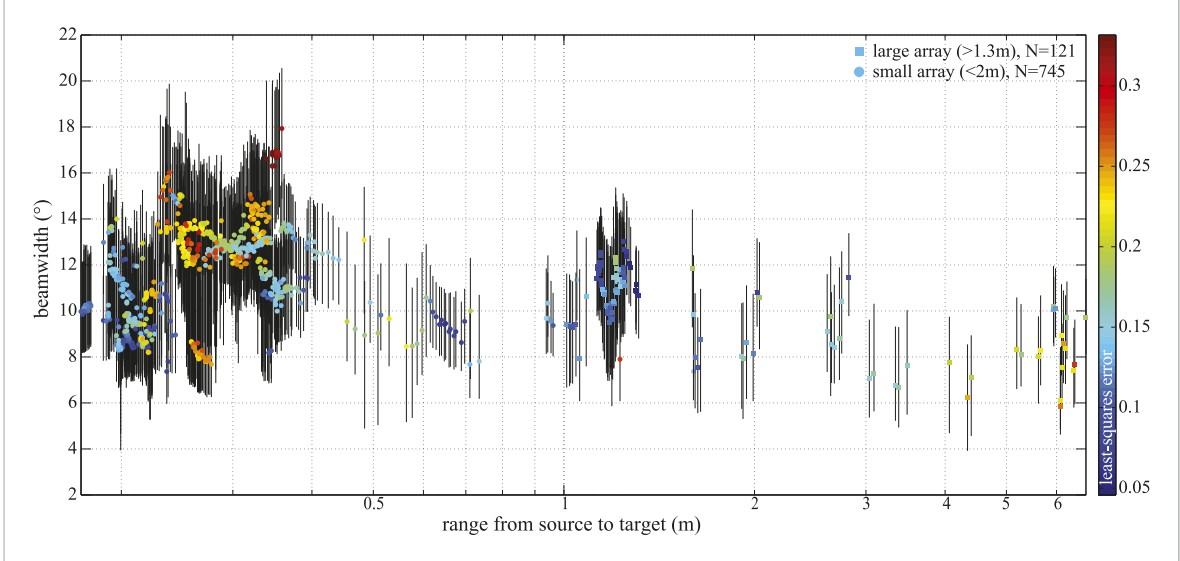

**Figure 4**. −3 dB beamwidth variation with range. Colored markers indicate beamwidth estimates based on the best fitting piston aperture, while the black vertical lines show the spread around the best fit (see lower panels in *Figure 4—figure supplement 1*). Data were gathered with star-shaped arrays in two configurations (*Figure 3C* and *Figure 4—figure supplement 1*): large (squares, N = 121) and small (circles, N = 745). Color indicates the least-square error associated with the fits. Only fits with error <0.2 were considered in the final analysis and presented in *Figure 3*. Distance from the sound source to the tip of the animal's rostrum was 17 cm.

The following figure supplement is available for figure 4:

**Figure supplement 1**. Array configurations and data fitting in experiment two.

operating (*Surlykke et al., 2009*), and the task at hand (*Jakobsen and Surlykke, 2010*), changes in their FOV during the buzz are accounted for by a concurrent drop in signal frequency content. Thus in bats, changes in FOV, emission rate, and signal frequency content during the buzz appear to be tightly interconnected (*Ratcliffe et al., 2013*). Our results show that in whales these parameters are independently controlled (*Figure 6* and *Figure 6—figure supplement 1*). Having a FOV that can be modulated independently of signal emission rate or frequency content (the latter often being dependent on the amplitude of the signal [*Au et al., 1995*; *Finneran et al., 2014*]) may be essential for managing flow of sensory information and optimizing long duration close-range prey tracking in acoustic scenes of varying complexity (*Figure 2*). The broad beam is advantageous to the porpoises at close range where it would reduce the likelihood of prey escaping perpendicularly to the approaching porpoise by vanishing from its acoustic FOV.

These findings support the hypothesis that porpoises dynamically control their acoustic FOV while tracking prey, and do so by altering the effective size of their radiating aperture. The mechanism underlying these adjustments may be muscle-induced phonic lips repositioning (*Cranford et al., 2014*), and melon and air sac deformations (*Moore et al., 2008*; *Huggenberger et al., 2009*). All toothed whales studied to date have similar facial musculature surrounding the melon (*Cranford et al., 1996*; *Harper et al., 2008*). All toothed whales then presumably have the ability to modify the melon's shape. Given the greater beam plasticity offered by this mechanism, compared to modulating the frequency content of clicks, we suggest that all toothed whales may be able to shape and modulate their beam this way.

Despite the independent evolution and very different means of sound generation and transmission, whales and bats have both evolved mechanisms to change their acoustic FOV while tracking prey. This suggests that beam plasticity has been a key driver in the evolution of echolocation, beyond simple orientation, for improved foraging success. Our results from these small toothed whales suggest that the demands of tracking moving prey over variable distances in complex acoustic environments have favored the evolution of a more sophisticated adjustment mechanism, in which pulse rate and beamwidth can be controlled independently. Compared to bats, the greater dynamic beam plasticity

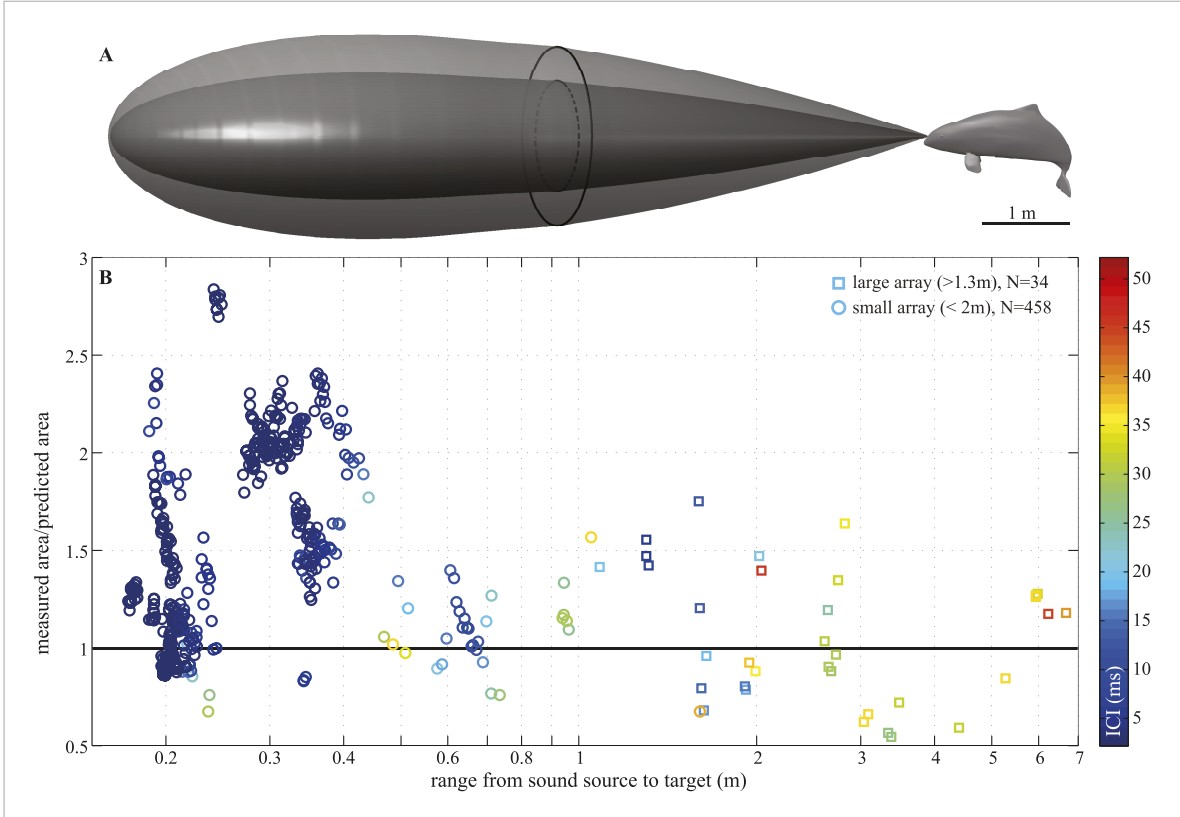

**Figure 5**. Beam adjustments can triple the ensonified area. (**A**) Approximate detection volume for a harbour porpoise tracking fish in a quiet environment, based on the active sonar equation (1) and source energy levels as measured. Fixed detection threshold (*Kastelein et al., 1999*) of 27 dB re 1 µPa²s and target strength of −36 dB for the Atlantic cod of 29–30 cm (*Au et al., 2007*) is assumed. Pattern of the outer beam (solid line cross section) is based on beamwidth estimates obtained at short target ranges. The inner, narrow beam (dashed line) is based on the directionalities measured at long range, representing predicted beam pattern had the porpoise not switched to 'wide-angle view'. (**B**) Relative change in the size of ensonified area ahead of the porpoise as it approaches a target. Surface area was computed as base of a cone with height equal to target range and an opening angle corresponding to the measured −3 dB beam angle (measured area, solid line in **A**) or median −3 dB beam angle calculated for long ranges (>2 m; predicted area, dashed line in **A**). Color indicates inter-click intervals. Squares and circles mark data points obtained with the large (N = 34) and small (N = 458) array, respectively. The bold horizontal line indicates points where the measured and the predicted areas are equal.

we have observed here likely reflects different sensory and ecological constraints. We propose that dynamic control of acoustic FOV in whales is a mechanism for the inclusion and exclusion of potential sensory information that allows these predators to quickly and repeatedly adjust to changes in habitat and prey trajectories.

## Materials and methods

All experiments were conducted in a semi-natural outdoor enclosure at Fjord&Bælt, situated in Kerteminde harbour, Denmark. The enclosure (approximately 34 × 17 m, natural sandy bottom at 3–5 m depth) is fenced off by a concrete wall alongshore, and nets on the two shorter ends. The net pen complex comprises two net-separated pools that allow for isolation of single animals for experimental work. All acoustic recordings in the present study were made in the smaller, 8 × 12 m net pen.

At the time the study was undertaken, the facility housed four harbour porpoises, three of which participated in the experiments: Freja (female, at Fjord&Bælt since April 1997, estimated to be 1–2 years old at arrival [*Lockyer, 2003*]), Eigil (male, at Fjord&Bælt since April 1997, estimated to be 1–2 years old at arrival [*Lockyer, 2003*]) and Sif (female, at Fjord&Bælt since July 2004, estimated to be 1-year-old at arrival [*Lockyer, 2003*]). All animals had extensive previous experience in various echolocation experiments, from being stationed at a target (e.g., [*Beedholm and Miller, 2007*; *Koblitz et al., 2012*; *Linnenschmidt et al., 2012*]) to free-swimming (e.g., [*Verfuss et al., 2005*;

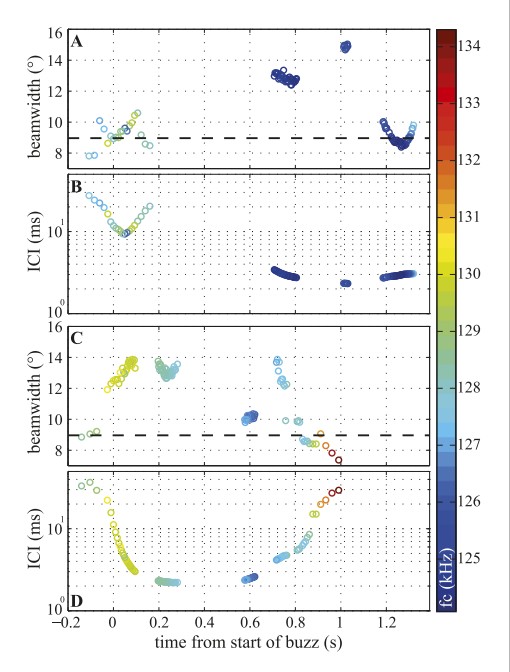

**Figure 6**. Temporal variation in beamwidth within the terminal buzz. Beamwidth changes in terminal phases of two trials (**A**, **C**) and their respective inter-click intervals (ICI; **B**, **D**). Color-coding represents centroid frequencies (fc) of signals. Dashed line in (**A**, **C**) corresponds to median beamwidth at long ranges (>2 m). The porpoise used different beamwidths whilst maintaining ICIs and vice versa. Both trials were recorded with the small star-shaped array (light blue in **Figure 3C**), but during the trial shown in (**C**, **D**) the porpoise was not blindfolded. Only data for clicks fulfilling inclusion criterion are presented.

The following figure supplement is available for figure 6:

**Figure supplement 1**. Frequency variation alone cannot explain the observed beamwidth changes.

Beedholm and Miller, 2007; DeRuiter et al., 2009; Wisniewska et al., 2012]), as well as carrying a tag (DeRuiter et al., 2009; Wisniewska et al., 2012). The animals were trained to participate in the experiments using operant conditioning and positive reinforcement (Ramirez, 1999).

## Experiment one

We recorded echolocation clicks from three blindfolded (i.e., wearing opaque silicone eye-cups) porpoises as they swam alone in the 3–4 m deep net-pen across the long side of the pool to capture dead, freshly thawed fish and towards a horizontal linear array of 8 calibrated Reson TC4014 hydrophones spaced 60 cm apart (Figure 3A and Video 1). The array was deployed at a depth of 75 cm and the fish were introduced approx. 3 m away from its centre, giving the array an effective angular resolution of ~12° at ranges of target interception (EAR = atan[0.6 m/3 m]). Signals were amplified and filtered using a custom-made conditioning box and then simultaneously A/D converted with 16-bit resolution at 500 kHz per channel (National Instruments PXI-6123, Austin, TX). All trials were monitored using a video camera (Profiline CTV7040, Abus, Germany) synchronized with the audio recordings.

Analyses were performed using Matlab (Math-Works, Natick, MA). We localized the animal at the time of each emission using hydrophone arrival-time differences (Madsen and Wahlberg, 2007). Porpoise positions were then verified with the synchronized videos. For each click, the time of maximum sound pressure on each hydrophone was identified, and the energy density of the signal was measured using a window of 30 µs before and 90 µs after the peak of the signal envelope. Such a window corresponds to the duration of a typical porpoise signal. Assuming spherical spreading, the apparent source level (ASL) (Madsen and Wahlberg, 2007; Finneran et al., 2014) was calculated from the recorded level (computed as energy flux density in a fixed window [peak-30 µs—peak +90 µs]) and the range. For a click to be included in the analysis its ratio of signal energy to the immediately preceding noise energy had to exceed 6 dB on all channels, and the maximum ASL should not have occurred on one of the outermost hydrophones. Radiation plots were created by plotting the ASLs against their respective angles relative to the estimated on-axis direction. First, the peak amplitude and angle were adjusted by interpolating between the peak ASL and the ASLs from the two neighbouring hydrophones using Lagrange interpolation (Menne and Hackbarth, 1986). All off-axis levels were then plotted as a function of the off-axis angle. The resulting transmission beam pattern was interpolated to a grid of 0.1°. The circular piston model was used to estimate the directivity index and −3 dB beamwidth of the beam pattern as described in (Møhl et al., 2003).

## Experiment two

A single harbour porpoise (Freja) was trained to swim across the short side of the pool and close in on a 50.8 mm-diameter spherical aluminum target, suspended by a nylon line just in front (5–40 cm,

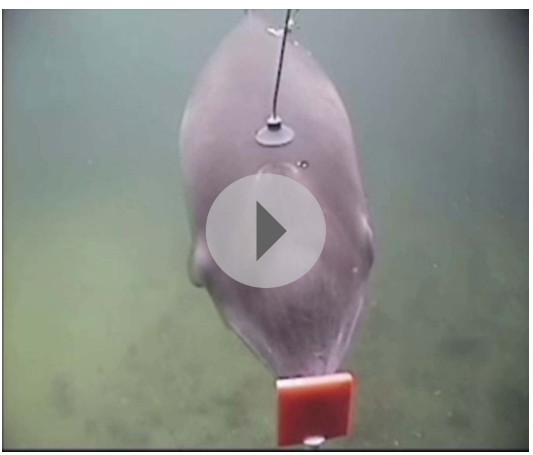

**Video 3.** Harbour porpoises can manipulate their melon while producing clicks. Video shows a harbour porpoise emitting echolocation click trains during a hearing test. It has been slowed down by a factor of two and synchronized with the output of a porpoise click detector. The porpoise depresses the melon as it switches to high repetition rate click trains. Conformation changes in the nasal complex can modulate the degree of sound collimation in the whale's forehead (*[Harper et al., 2008*; *Moore et al., 2008*; *Huggenberger et al., 2009*]) to change the field of view. Courtesy of Lee Miller.

depending on array configuration, see below and *Figure 3*) of the center of an array of 48 small (25 × 10 mm), custom-built hydrophones (*Wisniewska et al., 2012*) (*Video 2*). The hydrophones were attached to a mesh of 1 mm-diameter Dyneema string stretched over a 3 × 3.5 m (height by width) metal frame to form a grid of 5 × 5 cm squares. The hydrophones had a flat (±2 dB) frequency response between 100–160 kHz and were connected to a 48 channel conditioning box with 40 dB gain and a fourth order bandpass filter with −3 dB frequencies at 2 kHz and 200 kHz. The hydrophones were sampled continuously during the trials with 16-bit resolution at 500 kHz/channel by three synchronized National Instruments PXIE-6358 boards and streamed to disk, using custom made software developed in LabVIEW, National Instruments (*Source code 1*).

Differences in array sensitivity due to hydrophone arrangement and attachment were measured after each data collection and corrected during post-processing. The hydrophones were arranged in two star-shaped configurations (*Figure 3C* and *Figure 4—figure supplement 1*). To map the beam extent at long ranges (1.3–7 m), we used hydrophone spacing increasing toward the edge hydrophones 1.05–1.13 m from the centre. Consequently, along the vertical and horizontal axes the hydrophones were separated by: 5, 10, 15, 20, 25, and 30 cm. Along the diagonal axes the hydrophones were separated by 14.1, 14.1, 21.2, 28.3, and 35.4 cm. With hydrophones 14 cm apart, the effective angular resolution was ~6° at 1.3 m from the array's centre. To maintain high spatial resolution at short ranges (≤2 m), we displaced the target outward to 0.4 m from the array and rearranged the hydrophones, resulting in an array extending out to 0.5 m with hydrophones separated by 5, 5, 5, 15, and 25 cm along the vertical and horizontal axes, and by 7.1, 7.1, 14.1, and 21.2 cm along the diagonal axes. Thus, a sound source at 0.55 m from the array (i.e., the shortest range examined) could have had its beamwidth measured to within ~5°.

We pooled the two data sets together with a range overlap between 1.3–2 m. Data points acquired with the wider spacing when the animal was closer than 1.3 m were discarded, as were data acquired with the fine spacing when the porpoise was more than 2 m away. The porpoise was equipped with a DTAG-3 multi-sensor tag (*Johnson and Tyack, 2003*; *Johnson et al., 2004*; *Wisniewska et al., 2012*) attached with suction cups just behind the blowhole, to allow for measuring the range of the sound source to the target and the array from the difference in time-of-arrival of click–echo pairs. The tag sampled sound with 16-bit resolution at 500 kHz/channel and was synchronized with the array hydrophones (*Wisniewska et al., 2012*). In all but one (*Figure 6C,D*) of the analyzed trials, the porpoise was blindfolded with opaque eyecups.

All trials were monitored with a set of GoPro Hero 2 cameras (two on the heads of the trainers and one approximately 1.8 m behind the array; Eye of Mine Action Cameras, Carson, CA) synchronized with the DTAG-3 recordings.

The recorded trials were pre-screened for relatively straight approaches to the array using the videos. All subsequent sound analyses were performed using Matlab. Clicks from the study animal were identified in the DTAG-3 acoustic recordings using a supervised click detector. Spectral cues were used to eliminate occasional misdetections of echoes or signals from other porpoises in the neighbouring pen. Echograms were formed from the sounds recorded with the DTAG-3. Only trials with clear echoes from the target and the array were submitted to further analysis. Clicks from 13–21 key hydrophones of the large- and small star-shaped array were extracted using a supervised click detector with the synchronized timing of the clicks recorded on the animal as an input. Clicks from all

the verified channels were then combined into a single template and used for automatic click detection on the remaining channels. For each click, we identified a subset of channels with peak received levels exceeding the rms noise level of the channel by at least 14 dB. We calculated click energy in a fixed window (peak-30 µs—peak +90 µs) on each of the selected channels, fitted a surface to the values using the Matlab 'gridfit' function with grid spacing of 0.5 cm and determined the location of the beam axis as the peak of the fitted surface (*Figure 4—figure supplement 1*, upper panels). We ran a series of computer simulations of our methods applied to virtual sources of known piston sizes to evaluate the influence of (i) an animal's bearing in azimuth and elevation and (ii) beam axis displacement relative to the centre of the array on the beamwidth measurement error. Based on the results of these simulations, we restricted our analysis to only include clicks emitted when the porpoise was swimming directly toward the target (within ±15° vertically and horizontally from the array's centre) and with beam axis within (i) ±12 cm from the array centre for the large array, (ii) ±8 cm for clicks recorded with the small array and produced up to 1 s before the start of buzz, and (iii) ±6 cm for clicks made 1 s before buzz and onwards until the end of the target approach. The estimated error was thereby limited to ≤±0.5°.

The simulations also verified that at these displacements the location of the beam axis could be estimated with an accuracy of ±1.5 cm at the ranges considered in this study. We used hydrophone arrival-time differences to compute the animal's bearing (azimuth and elevation) to the centre of the array.

For each click, we followed the method of *Kyhn et al. (2010)* to fit the energy levels received at the hydrophones to the beam pattern of a circular piston that fulfilled the following criteria: (i) it was at the same range to the array as the porpoise emitting the click, (ii) it was centered on the estimated beam axis and (iii) it transmitted the click recorded on the hydrophone closest to that axis. Given that the orientation of the porpoise relative to the array was constantly changing as the animal was approaching the target (*Video 2*), the beam pattern was assumed to be rotationally symmetrical around the acoustic axis. Furthermore, this assumption allowed us to utilize information from all hydrophones fulfilling the signal-to-noise ratio criterion to find the best fitting aperture. We carried out a Monte Carlo simulation using theoretical piston transducers with diameters of 1/3 to 3 times 8.3 cm (i.e., the best fitting vertical equivalent aperture in [*Koblitz et al., 2012*]) in 0.1 cm increments, and the circular piston model of *Au et al. (1987)*. The diameter of the piston that matched the data best was found by means of a non-linear least-squares method (see *Figure 4—figure supplement 1*). Only fits with $R^2 > 0.8$ (*Figure 4*) were kept for final analysis, from which the −3 dB beamwidth and the equivalent piston radius were extracted.

To examine the relative change in the size of ensonified area ahead of the porpoise as it approached a target, we computed the surface area as base of a cone with height equal to target range and an opening angle corresponding to (i) the measured −3 dB beam angle and (ii) median −3 dB beam angle calculated for long ranges.

We pooled beamwidth and distance to target data for all clicks from Experiment 1 for the three porpoises and ran a regression analysis on these data. We then pooled beamwidth and distance to target data for both array configurations from Experiment 2 and ran a regression analysis on these data as well. Additionally, we used hierarchical cluster analysis (centroid, non-standardized) to assign click beamwidth data from Experiment 2 to one of three clusters and compared these clusters using ANOVA with respect to click distance to target. All statistical analyses were carried out using JMP v. 11.2 (SAS Institute, Cary, NC, USA).

## Live prey capture

To explore the acoustic scene experienced by a toothed whale tracking active prey, we deployed DTAG-3 tags (with the same recording settings as in experiment two) on porpoises involved in pursuit of small (~15 cm), live trout in the pen complex of Fjord&Belt, where the animals have access to a natural sandy bottom at 3–4 m depth. This unique setting approximates what this shallow-water predator might encounter in its natural surroundings. The tags recorded the whale's echolocation clicks as well as echoes from the fish and other objects and surfaces in the whale's surroundings (e.g., water surface and sea floor). The recorded sounds were used to form stack plots, or echograms, of sound envelopes synchronized to the outgoing click as in echosounder images (*Johnson et al., 2004*). These allowed us to follow movements of the echolocator and its prey in the environment (*Figure 2*). The time delay between the echolocation click and the echo, multiplied by one-half of the sound

speed (1500 m/s was assumed) gives the distance to the target. Delays to the surface and bottom echoes approximate the animal's depth and altitude above the sea floor, respectively. Time delays between different echo groups represent their relative proximity.

## Anatomy of a porpoise head

A video of melon deformations can be found online (*Video 3*). To visualize the anatomy of the head, a dead specimen was scanned in a 1.5T Siemens Avanto MRI system (Siemens Medical Solutions, Germany). A Flash 3D T1 weighted pulse-sequence with the following parameters was used: TR 14.8 ms, TE 3.38 ms, $\alpha = 15°$, NEX = 3, spatial resolution = $0.64 \times 0.64 \times 0.75$ mm. Following acquisition, segmentation and modeling were done using Amira 5.3.3 (Visualization Science Group, Germany).

## Acknowledgements

We are grateful to JH Kristensen, JD Hansen, S Hansen, C Eriksson, M Dyndo, and L Jacobsen and the staff at Fjord&Belt for assistance with data collection, and H Lauridsen at Skejby University Hospital for help with MRI scanning. We thank NU Kristiansen, JS Jensen, M Dyndo, and K Ydesen for helping with construction of the recording setup, H-U Schnitzler (Universität Tübingen) for lending recording gear, and T Hurst (Woods Hole Oceanographic Institution) for providing the DTAG-3. WWL Au, AH Bass, MB Fenton, A Surlykke, and L Wiegrebe kindly provided comments that improved the manuscript. The animals are maintained by Fjord&Belt, Denmark, under permits no. SN 343/FY-0014 and 1996-3446-0021 from the Danish Forest and Nature Agency.

## Additional information

### Funding

| Funder | Author |
| --- | --- |
| Oticon Foundation Denmar | Danuta M Wisniewska, Christian B Christensen |
| Det Frie Forskningsråd | John M Ratcliffe, Mark Johnson, Peter T Madsen |
| National Instruments | Danuta M Wisniewska, Kristian Beedholm, Peter T Madsen |

The funders had no role in study design, data collection and interpretation, or the decision to submit the work for publication.

### Author contributions

DMW, Designed the experiments, carried out the large array recordings for experiment two, conducted the live prey capture trials, analysed and interpreted the data, wrote the manuscript, Conception and design, Acquisition of data, Analysis and interpretation of data, Drafting or revising the article; JMR, Conducted experiment one, contributed to the design of experiments, wrote the manuscript, Conception and design, Acquisition of data, Drafting or revising the article; KB, Designed the experiments, wrote the data acquisition software, analysed and interpreted the data, revised the manuscript, Conception and design, Acquisition of data, Analysis and interpretation of data, Drafting or revising the article; CBC, Analyzed the scan data and contributed to the data interpretation, revised the manuscript, Analysis and interpretation of data, Drafting or revising the article; MJ, Conducted the live prey capture trials, contributed analytical tools, contributed to interpretation of data and revised the manuscript, Acquisition of data, Analysis and interpretation of data, Drafting or revising the article; JCK, Conducted experiment one, analysed the data and revised the manuscript, Acquisition of data, Analysis and interpretation of data, Drafting or revising the article; MW, Contributed to the design of the experiments, conducted experiment one, analysed the data and revised the manuscript, Conception and design, Acquisition of data, Analysis and interpretation of data, Drafting or revising the article; PTM, Designed the experiments, conducted the live prey capture trials, contributed to the analysis and interpretation of data, wrote the manuscript, Conception and design, Acquisition of data, Analysis and interpretation of data, Drafting or revising the article

#### Author ORCIDs
Danuta M Wisniewska, http://orcid.org/0000-0002-3599-7440

#### Ethics
Animal experimentation: The animals are maintained by Fjord&Belt, Denmark, under permits no. SN 343/FY-0014 from the Danish Ministry of Food, Agriculture and Fisheries, and 1996-3446-0021 from the Danish Forest and Nature Agency (under the Danish Ministry of the Environment). Their care and all experiments are in strict accordance with the recommendations of the Danish Ministry of Food, Agriculture and Fisheries (issuing the permit to keep the animals), the Danish Ministry of the Environment (permit for catching the animals) and the Danish Council for Experiments on Animals (always contacted for permits when appropriate—but in the case of this study such permit was not required).

## Additional files

### Supplementary files
• Source code 1. LabVIEW (2012 version) source code files for the National Instruments PXIE-6358 system, sampling 48 AD channels at 500 kHz/channel. An independent multifunction USB device (National Instruments USB-6251) delivered a trigger pulse for the PXIE system alongside a high-frequency pulse (a frequency-modulated sweep) that was transmitted into the water to synchronize the hydrophone and DTAG-3 recordings. The main VI is 'use_this_sampler_for_now03.vi'. Supporting VIs should be kept in the same folder. This software was built for a specific PXIE system, and modifications will be necessary for it to run other systems. Any questions regarding this should be addressed to kristian.beedholm@bios.au.dk.

### Major dataset
The following dataset was generated:

| Author(s) | Year | Dataset title | Dataset ID and/or URL | Database, license, and accessibility information |
|---|---|---|---|---|
| Wisniewska DM, Ratcliffe JM, Beedholm K, Christensen CB, Johnson M, Koblitz JC, Wahlberg M, Madsen PT | 2014 | Data from: Range-dependent flexibility in the acoustic field of view of echolocating whales | 10.5061/dryad.11b0f | Available at Dryad Digital Repository under a CC0 Public Domain Dedication. |
| Wisniewska DM, Ratcliffe JM, Beedholm K, Christensen CB, Johnson M, Koblitz JC, Wahlberg M, Madsen PT | 2015 | Data from: Range-dependent flexibility in the acoustic field of view of echolocating whales | 10.5281/zenodo.17195 | Full data available at Zenodo under a CC0 Public Domain Dedication. |

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
