## [Decision Letter]

Thank you for sending your work entitled “Range-dependent flexibility in the acoustic field of view of echolocating whales” for consideration at *eLife*. Your article has been favorably evaluated by Eve Marder (Senior editor), a Reviewing editor, and three reviewers.

The Reviewing editor and the reviewers discussed their comments before we reached this decision, and the Reviewing editor has assembled the following comments to help you prepare a revised submission.

Comments for authors:

1) Title: Suggestion to change the title for clarity. Possible title: “Range dependent flexibility in the acoustic field of view of echolocating porpoises.” You might include the species name in the title as well.

2) Abstract: The Abstract appears contradictory in stating that “porpoises… unlike echolocating bats, are able to change beamwidth within” the terminal phase of attack, but also that “both whales and bats have evolved mechanisms to change their FOV… for prey tracking.” You need to be more precise if you want to argue that they have converged but are different. In addition, you state “that harbor porpoises broaden their biosonar beam by >50% in the terminal phase”, but that is not a fair summary of the results, showing very little broadening on average. You should say “harbor porpoises can broaden their…”, to report the results objectively.

3) Text:

a) Please provide more information about the animals used including previous experience with echolocation experiments.

b) Explain why you used only a single animal for the second two experiments and analyzed only a small subsample of the data. Please be clearer about the possible effects of this selective process. Was the performance of their subject exceptional? Did they analyze all the analyzable click trains, or only a subset? Did their criterion reflect a behavioral difference (e.g., the data produced by one type of approach to the target being easier to analyze than another type of approach). Since this phenomenon has not been described previously would be inter2preted differently if the animals used a variety of strategies to home in on the target.

c) The MRI images of the structure of the melon and phonic lips are not described in sufficient detail. Please provide a brief description of how these images compare with previous images of this type from harbor porpoises. Is this the first time simulations of production have been done for the harbor porpoise, or were there others? Were they consistent?

d) There is a discrepancy between the text describing “a dramatic broadening of the beam” and the actual data showing almost no effect, when comparing the median beam width. However, is median the best measure? Figure 4 indicates a strong tendency to broaden the beam in the last phase. Wouldn't this be better described by another measure than median? The data are probably not Gaussian, but even so, the average (or “weight”) might illustrate the effect.

e) How were the data pooled to determine medians and distribution? According to array type? Distance? ICI?

f) Please present a more detailed data taking protocol (individual porpoise, distance between target and array, array type and deployment place) into consideration. Figure 5 shows the results from two very different trials suggesting that pooling the data might mask some of the effects. Figure 3–figure supplement 2 show many data at ca. 1.2 m range. Was there anything specific here? In addition, this figure (which should be included instead of e.g. Figure 2, which is not really used and mostly repeats the message from earlier similar echograms) suggests a very interesting pattern with broadest beam width at between 25 and 35 cm target range.

g) Did individual porpoises show the same results? How did the data from the two arrays compare for the porpoise investigated in both series?

h) What is portrayed in Figure 3–figure supplement 3? If the target was centered on the 0,0 point and the vertical axis was the Y-axis, it can be puzzled out but it should be easier. For example, show the location of the target, and say “bearing” and “azimuth” instead of showing two bearings.

i) At the end of the Introduction, the analysis provided does not directly address the hypothesis to be tested, but perhaps this can possibly be corrected by reanalysis of the existing data, and that this could be done in a reasonable time. The most important result described in the Abstract is that “porpoises broaden their sonar beam by >50%” while the results of experiment 2 state: “the porpoise could almost double its beamwidth at short target ranges (Figure 3) from a median half-power beamwidth of 9° (6.6-11.8) to a maximum of 15° (median = 10.8, min = 7.4).” So, is the difference 50%, as the Abstract says, or 200% as the Results say? I assume that this does not refer to experiment 1, as the authors state “the relatively large hydrophone spacing in this experiment… prevented us from drawing strong conclusions about the exact extent of beam widening.” The comparison in the Results of experiment 2 is inexplicably between a median and a maximum. If you compare the two median values, 9° (6.6-11.8) to 10.8° (7.4-15), I am not at all convinced that there even is a significant difference. If the paper wants to reach conclusions about changes in beamwidth with distance to target or stages of capture, then it needs a valid statistical comparison of the two distributions. Since the goal is to test whether the porpoise changes FOV adaptively during prey interception, then I think the test should be organized using each approach as the basic unit of analysis. Figure 3 merges data from different approaches, and even totally different receiver set-ups. The actual analysis of adaptive changing of FOV during interception should only test for changes in FOV within each approach. The prediction should be that beamwidth expands as porpoise closes on prey.

j) Do porpoises change beamwidth independent of frequency by changing the configuration of the melon and air sacs? The paper several times cites Figure 5 as demonstrating this. While it shows two examples relating ICI, bandwidth, and center frequency in the context of time from start of buzz, this figure does not actually show a clear analysis that beamwidth is independent of frequency. This requires a statistical test of this point, perhaps against a physical model of how beamwidth would change with fc for a simple static transducer. A plot of beamwidth against fc would also help.

Specific linguistic concerns:

Introduction:

This is a bit picky, but beaked whales only decrease intensity as they switch from search to buzz; most of their closing on prey is not accompanied by decrease. There are very limited data on odontocete species with auditory gain control.

I think you need to walk the reader through the echogram. I think only a tiny minority would be able to link what you say in the text to what you see in Figure 2 without help. Where was this recorded? In the pool? What experiment? The figure is beautiful, but the data in this figure (actual distances to bottom and surface) are not used for anything in the Discussion.

Could the large increase in the experiment with linear array be due to the porpoise not focusing on the array (vertically) before closing in?

Results:

How did you control for beam axis in the vertical plane?

I suggest not reporting a difference from 9° to 18° as 99% difference. “100% different” indicate “no overlap”, and accordingly 99% would suggest “almost completely different”. Doubling the sonar beam width would be less confusing.

It is not easy to extract the real results when you provide a comparison between the median (at longer ranges) and the one maximum value at short range. If you want to “break records” it would be fairer to refer to a minimum value vs. a maximum value. Then the reader would know you are referring to the extremes of the dataset.

What does “better approximated” mean precisely?

Please explain which numbers you took to calculate the “at least three times increase”. Figure 5, lower part, seems to indicate that the beam can also be very broad before the buzz.

Discussion:

It is hard to see how the different array configurations could have created such a big difference. But a possible correlation to behavior/context could perhaps: either the freely floating fish with the linear array or the large clutter wall from the star-shaped array. It appears easy to test.

I think that the relation between wavelength and size of the lens/melon differs so radically that this analogy is probably not correct in terms of the physics.

I do not believe that the pupil and lens are not able to change FOV as much as you discuss here for echolocation.

Materials and methods:

How can you accurately interpolate to 0.1° if EAR is only good to 12°?

What does “just in front” mean? Few cm? mm? Figure 3 indicates that the target can be in two positions: very close to the center of the array and 1 (?) m in front of it? Figure 3, not clear.

How did arrangement of the array change the sensitivity of the hydrophones?

Is vertical axis tested?

The subsection headed “Live prey capture”: Is this dataset (and Figure 2) relevant for the study? Apparently, the data are not used for anything. The relevance would be obvious if both the array and the tag had recorded a trial in order to reveal possible correlation between changes in the echogram/acoustic scene and beam width, but that is not the case, as far as I can see.

Do you really think Atlantic cod is a good TS for porpoise prey? Most would be much smaller and lower TS.

Figure 5 is very interesting. The two trials are surprisingly different. Why is that? Were these recorded with the linear array or the star shaped? Probably star shaped, but what configuration?

Video 3: The video shows depression in the whole buzz phase. If that is a proxy for broadening the view, it suggests a constant broad beam in the buzz and not variation as suggested by Results and Discussion in this manuscript?

Please clarify what are the supplementary figures reporting and why.

---

## [Author Response]

*1) Title: Suggestion to change the title for clarity. Possible title:* “*Range dependent flexibility in the acoustic field of view of echolocating porpoises.*” *You might include the species name in the title as well*.

Good point, thank you for this suggestion. The title has now been changed accordingly.

*2) Abstract: The Abstract appears contradictory in stating that* “*porpoises… unlike echolocating bats, are able to change beamwidth within*” *the terminal phase of attack, but also that* “*both whales and bats have evolved mechanisms to change their FOV… for prey tracking.*” *You need to be more precise if you want to argue that they have converged but are different. In addition, you state* “*that harbor porpoises broaden their biosonar beam by >50% in the terminal phase*”*, but that is not a fair summary of the results, showing very little broadening on average. You should say* “*harbor porpoises can broaden their…*”*, to report the results objectively*.

Again, fair points, thank you. We have changed the Abstract accordingly.

*3) Text*:

*a) Please provide more information about the animals used including previous experience with echolocation experiments*.

Thank you for this suggestion. We have now added more information on the porpoises’ age, sex and experience in echolocation experiments.

*b) Explain why you used only a single animal for the second two experiments and analyzed only a small subsample of the data. Please be clearer about the possible effects of this selective process. Was the performance of their subject exceptional? Did they analyze all the analyzable click trains, or only a subset? Did their criterion reflect a behavioral difference (e.g., the data produced by one type of approach to the target being easier to analyze than another type of approach). Since this phenomenon has not been described previously would be interpreted differently if the animals used a variety of strategies to home in on the target*.

At the time the large-array experiments were conducted, Eigil, the male porpoise, was involved in an intensive training program for a psychophysical study using a “go-no go” procedure (see Linnenschmidt et al. 2012 and Linnenschmidt et al. 2013 for details). He was thus not available for our study.

Initially, we did, however, record both of the two females: Freja and Sif. Unfortunately, Sif became ill, and her training time had to be cut down and she never learned the task properly.

The 3 porpoises kept at Fjord & Bœlt also undergo daily husbandry training and are used in public presentations. Hence, training time and the amount of fish the animals were fed per day limited the number of porpoises and time assigned to our study.

With respect to the last point, we did analyze all analyzable click trains, using the standards we set for data quality. We feel our results can be extrapolated to harbor porpoises in general because, one, the results from the first experiment included all 3 porpoises (and all behaved similarly) and, two, these porpoises behave normally in the context of other echolocation parameters reported on from porpoises at other research centers and in the wild.

*c) The MRI images of the structure of the melon and phonic lips are not described in sufficient detail. Please provide a brief description of how these images compare with previous images of this type from harbor porpoises. Is this the first time simulations of production have been done for the harbor porpoise, or were there others? Were they consistent*?

We agree, and we have therefore expanded on the section that presents the scanning data. In that section we also compare in more detail with the one entirely anatomical study done on the porpoise head by Huggenberger and colleagues in 2009. Sound production has not been modelled in porpoises using scanning data, but interestingly Dr. Huggenberger concludes his anatomical paper by stating the following on the context of the musculature and the melon:

“Furthermore, the intermedius muscle (im) may pull the dorsal part of the melon caudally (Figure 9) and rostral parts of the anterointernus muscle (ai) may pull the melon terminus ventrally, thus changing its height. However, it would be a matter of speculation to decide how much such a change in shape can contribute to a potential modulation of the sonar beam.”

Here, we confirm for the first time that these muscles can modulate the shape of the melon, and that the beam width is indeed very adaptable in these animals. Thus, no one has previously modelled this, but this earlier study predicted that beams may be modulated by changing the melon confirmation. We have highlighted that fact in the revised manuscript.

*d) There is a discrepancy between the text describing* “*a dramatic broadening of the beam*” *and the actual data showing almost no effect, when comparing the median beam width. However, is median the best measure?*
Figure 4
*indicates a strong tendency to broaden the beam in the last phase. Wouldn't this be better described by another measure than median? The data are probably not Gaussian, but even so, the average (or* “*weight*”*) might illustrate the effect*.

Agreed. We have now used mean, rather than median. We have also removed “dramatic” from the text (it appeared once in the Discussion).

*e) How were the data pooled to determine medians and distribution? According to array type? Distance? ICI*?

Data from the two experiments were examined separately. For the large star-shaped array in experiment two, we measured beamwidth for all clicks emitted at source ranges greater than 1.3 m from the array. Ranges ≤2m were examined with the small star-shaped array. These data were then pooled together (i.e. there was a range overlap of the two arrays at 1.3-2 m). For the summary statistics, data from the two experiments were divided into two long- and short-range clicks based on source-target range (≤2m vs >2m).

However, we agree that pooling by ICI is a better approach. For each experiment, we have, therefore, now divided the data into buzz clicks (inter-click intervals ≤13 ms) and regular clicks (inter-click intervals >13 ms; buzz onset threshold was based on ICI distributions examined in Wisniewska et al. 2012 for the same porpoises). The distributions of beamwidth in the two groups have been added to Figure 3.

*f) Please present a more detailed data taking protocol (individual porpoise, distance between target and array, array type and deployment place) into consideration.*
Figure 5
*shows the results from two very different trials suggesting that pooling the data might mask some of the effects. Figure 3–figure supplement 2 show many data at ca. 1.2 m range. Was there anything specific here? In addition, this figure (which should be included instead of e.g.*
Figure 2*, which is not really used and mostly repeats the message from earlier similar echograms) suggests a very interesting pattern with broadest beam width at between 25 and 35 cm target range*.

Trials with Freja, the porpoise used in experiment 2, were quite stereotyped; array location and distance between the target and the array did not change between the trials with a given array configuration. But in the trial presented in Figure 5, the porpoise was not blindfolded. More information about the animals, target ranges and blindfolds has been added to the manuscript text and figure legends. See Materials and methods, second paragraph and Figures 3 and 5.

Figure 3–figure supplement 2 has now been included in the manuscript as a standalone Figure 4. Toothed whales tend to initiate buzz when about a body length away from the intended target (and perhaps at even shorter ranges for stationary man-made targets (see e.g. Wisniewska et al. 2012)). Therefore, there will naturally be more clicks from that range onwards. While approaching a target, the animals scan their beam back and forth over the target, which, if the animal’s behaviour is very stereotyped, could lead to “click gaps” at some ranges. We are not sure why there is an accumulation of data points at 1.2 m, but we can speculate that that could have been the preferred range of transition to a buzz, during which the porpoise might have been more likely to focus its beam on the target. The fact that the data are from a single individual may have further exaggerated the effect.

Given that *eLife* does not limit the number of figures, we would like to keep Figure 2 in the manuscript. Please see our detailed justification below.

*g) Did individual porpoises show the same results? How did the data from the two arrays compare for the porpoise investigated in both series*?

Yes, there were some individual differences, but it is hard to address them with two trials per animal. The most extreme beamwidth values were all recorded from Eigil. Freja, the porpoise participating in the second experiment, broadened her beam up to a maximum of 24.3° when capturing fish in experiment one (as opposed to a maximum of 15.1° in experiment 2). We have now used separate symbols to represent data points from the different animals in experiment one.

*h) What is portrayed in Figure 3–figure supplement 3? If the target was centered on the 0,0 point and the vertical axis was the Y-axis, it can be puzzled out but it should be easier. For example, show the location of the target, and say* “*bearing*” *and* “*azimuth*” *instead of showing two bearings*.

Good point, but given the comment, the figure has been removed altogether.

*i) At the end of the Introduction, the analysis provided does not directly address the hypothesis to be tested, but perhaps this can possibly be corrected by reanalysis of the existing data, and that this could be done in a reasonable time. The most important result described in the Abstract is that* “*porpoises broaden their sonar beam by >50%*” *while the results of experiment 2 state:* “*the porpoise could almost double its beamwidth at short target ranges (*Figure 3*) from a median half-power beamwidth of 9° (6.6-11.8) to a maximum of 15° (median = 10.8, min = 7.4).*” *So, is the difference 50%, as the Abstract says, or 200% as the Results say? I assume that this does not refer to experiment 1, as the authors state* “*the relatively large hydrophone spacing in this experiment… prevented us from drawing strong conclusions about the exact extent of beam widening.*” *The comparison in the Results of experiment 2 is inexplicably between a median and a maximum. If you compare the two median values, 9° (6.6-11.8) to 10.8° (7.4-15), I am not at all convinced that there even is a significant difference. If the paper wants to reach conclusions about changes in beamwidth with distance to target or stages of capture, then it needs a valid statistical comparison of the two distributions. Since the goal is to test whether the porpoise changes FOV adaptively during prey interception, then I think the test should be organized using each approach as the basic unit of analysis.*
Figure 3
*merges data from different approaches, and even totally different receiver set-ups. The actual analysis of adaptive changing of FOV during interception should only test for changes in FOV within each approach. The prediction should be that beamwidth expands as porpoise closes on prey*.

The 50% referred to the change in beamwidth, while the 200% referred to the change in the ensonified cross-sectional area. In the new version of the manuscript, we have refrained from using % and are now providing the difference in beamwidth in degrees instead.

With respect to the hypothesis to be tested, we understand this important point of critique. However, we cannot reanalyse our data to address it. As the reviewer her/himself points out, we used different setups for long- and short-range recordings to cover the full extent of the beam at relatively long ranges and provide the necessary resolution at short ranges. Doing what the reviewer asks would require a setup with twice as many hydrophones, which with our current equipment, is not possible, given that we have to sample at 500 kHz per channel. Recording using 48 hydrophones simultaneously at these sampling rates, we were limited by our ability to stream the data to the computer.

Furthermore, given our strict click selection criteria (necessary to provide a reliable estimate of the beamwidth), even with a larger/more hydrophone-dense array, we are not sure if we could address this comment. We are not certain if the trials would provide enough data points to run tests on a trial-by-trial basis.

Following the reviewers’ advice, we have, however, performed a regression and hierarchical cluster analysis on the available beamwidth vs. range data and found that the distributions are indeed statistically different. The statistical results have been incorporated into the manuscript. In brief, they show that for experiment one, beamwidth and distance to target are inversely related. That is, that as the animals approach their targets, they broaden their sonar beam. We tested this using regression analysis.

We found the same significant inverse relationship for experiment 2, but the r2 value was much lower. Thus, we ran a combination of cluster analysis and ANOVA. We found that those clicks with the greatest beamwidth were produced significantly closer to the target than those clicks produced furthest away.

*j) Do porpoises change beamwidth independent of frequency by changing the configuration of the melon and air sacs? The paper several times cites*
Figure 5
*as demonstrating this. While it shows two examples relating ICI, bandwidth, and center frequency in the context of time from start of buzz, this figure does not actually show a clear analysis that beamwidth is independent of frequency. This requires a statistical test of this point, perhaps against a physical model of how beamwidth would change with fc for a simple static transducer. A plot of beamwidth against fc would also help*.

Thank you for this excellent suggestion. A plot of beamwidth against centroid frequency overlaying the results of a model of frequency-dependence of beamwidth for static circular piston transducers of different sizes has now been added to the manuscript (Figure 6—figure supplement 1). The plot shows that the variation of centroid frequencies measured in the two experiments could not explain the observed beamwidth changes. Even the best-fit models provided rather poor match to the data (RMSE=5.9 and SSE=2604 for experiment 1; RMSE=1.8 and SSE=1556 for experiment 2). To be sure, we ran exact Spearman's correlation tests of fc versus beamwidth, and found no significant relationship in the data from either experiment (experiment 1: P=0.76, experiment 2: P= 0.67). We have now added that information to the text.

*Specific linguistic concerns*:

Introduction:

*This is a bit picky, but beaked whales only decrease intensity as they switch from search to buzz; most of their closing on prey is not accompanied by decrease. There are very limited data on odontocete species with auditory gain control*.

We agree that it seems that odontocetes may show a whole spectrum of strategies of level adjustments to target range, dependent on the context of prey pursuit, and likely prey behaviour (see for example Wisniewska et al. 2014). While it was not our intention to state that all toothed whales and bats adjust their levels to fully compensate for the transmission loss, we can see how the sentence could be interpreted as such statement. We have therefore modified the sentence to clarify that.

*I think you need to walk the reader through the echogram. I think only a tiny minority would be able to link what you say in the text to what you see in*
Figure 2
*without help. Where was this recorded? In the pool? What experiment*?

Good point. All the information can be found in one of the subsections of the Material and methods section, but we agree that that was not necessarily clear in the figure legend or the main text. A reference to the subsection has now been added to the figure legend.

Could the large increase in the experiment with linear array be due to the porpoise not focusing on the array (vertically) before closing in?

We have no detailed control over the vertical direction of the beam using the linear array. The porpoise was approaching the linear array throughout the recording, and from the video observations there was no obvious change in the porpoise swimming direction when being close to as compared to being far away from the array (with respect to either head angle or the depth of the porpoise itself). Any possible effects of the porpoise being slightly off axis vertically during the linear array recordings were ruled out by the subsequent planar array measurements. There is a new sentence included in the text to clarify this.

Results:

How did you control for beam axis in the vertical plane?

We have applied on-axis criteria that in a large number of papers seem to work convincingly (e.g. Jensen et al. 2009; [32]; Wahlberg et al. 2011; Jensen et al. 2013). But no, we could not control for the vertical direction of the beam using a horizontal linear array, which is why the planar arrays (experiment 2) were necessary to get detailed measurements on the beam broadening. We have included a sentence to clarify this in the new version.

*I suggest not reporting a difference from 9° to 18° as 99% difference.* “*100% different*” *indicate* “*no overlap*”*, and accordingly 99% would suggest* “*almost completely different*”*. Doubling the sonar beam width would be less confusing*.

We have now pooled the data based on their ICI, rather than range, and so the results are different. But we agree that perhaps % difference is not the best approach. We are now providing the actual difference in degrees instead.

*It is not easy to extract the real results when you provide a comparison between the median (at longer ranges) and the one maximum value at short range. If you want to* “*break records*” *it would be fairer to refer to a minimum value vs. a maximum value. Then the reader would know you are referring to the extremes of the dataset*.

We have now used mean, rather than median. The minimum and maximum values are provided in the brackets for comparisons.

What does “better approximated” mean precisely?

As suggested earlier on this review, we now use mean rather than median values. We also agree that “better approximated” is vague, and have replaced this with more precise language which reflects the fact that the long range click data are more evenly and tightly spread around the mean beamwidth value and thus better described by the mean than are the short range data.

*Please explain which numbers you took to calculate the* “*at least three times increase*”*.*
Figure 5*, lower part, seems to indicate that the beam can also be very broad before the buzz*.

The sentence states “up to three times” and refers to the size of ensonified area ahead of the porpoise (which will depend on the beamwidth and the range to the target), rather than the beamwidth. We base that statement on the data presented in Figure 5 (Figure 4 in the previous version of the manuscript), where we show the ratio of area to area measured at a given target range to the area predicted for that range based on the median -3dB beam angle calculated for ranges >2m (i.e. predicted area had the porpoise not switched to “wide-angle view”).

Discussion:

*It is hard to see how the different array configurations could have created such a big difference. But a possible correlation to behavior/context could perhaps: either the freely floating fish with the linear array or the large clutter wall from the star-shaped array. It appears easy to test*.

We agree that the differences in array configurations are unlikely to generate such large differences in beamwidth as an artefact, and that it is the behaviour and/or context that likely have caused these differences. We have added a sentence pointing this out to the manuscript.

*I think that the relation between wavelength and size of the lens/melon differs so radically that this analogy is probably not correct in terms of the physics*.

We agree that the reviewers may be right, but argue that the analogy still has illustrative value (especially for those readers not familiar with whales/echolocation). We have, however, reworded this section to reflect that this analogy is speculative: specifically, we have removed the phrase “acoustic analogue” as this phrase did suggest an analogy argument based in physics.

I do not believe that the pupil and lens are not able to change FOV as much as you discuss here for echolocation.

A true, adjustable aperture to change the field of view can only be achieved in an optical system with at least two lenses, and no mammals or birds are equipped with two lenses to our knowledge. According to Professor Ronald Kröger at Lund University that we have consulted with on this topic, some shallow water fish have dorsal iris or skin flaps that seem to block out the sun when light adapted, while the upward direction is free in dark-adapted eyes. That is indeed an example of a change in FOV, but we have searched the literature and consulted people at the Lund vision group, and from that it is clear that a vertebrate generally cannot change its FOV, and no mammals have been shown to do it. What most animals can do is to move the FOV around by either moving the eyes or the head to effectively increase the search volume, but toothed whales can do that too by moving their head around. What we show here is that the acoustic FOV can be changed independent of head movements, which is not possible in mammalian and avian visual systems. We have clarified this point in the revised text.

Materials and methods:

How can you accurately interpolate to 0.1° if EAR is only good to 12°?

Details of the interpolation procedure have been included in the Materials and methods section. The 12° is only for the final clicks very close to the array – for the initial clicks further from the array the angular resolution were less than 4 degrees. This is now clarified in the beginning of the result section discussing the quality of the linear array data. It is true that in this experiment we have no control as to how accurate the interpolation is. We have data currently in press in Journal of Experimental Biology (Jensen et al.) using a transducer mimicking porpoise clicks and beam patterns and recorded with a linear array showing that with a similar array geometry and an angular resolution of 6° it is possible to achieve beam measurements with sub-degree precision, so using an interpolation step of 0.1° on the data of this manuscript does not seem that unrealistic.

*What does* “*just in front*” *mean? Few cm? mm?*
Figure 3
*indicates that the target can be in two positions: very close to the center of the array and 1 (?) m in front of it?*
Figure 3*, not clear*.

5-40 cm, depending on array configuration. We agree that this was not clearly stated in the original manuscript. Thank you for pointing this out. The information has been added to the text and the legend of Figure 3. The projections of the targets on the x-y plane in Figure 3) were meant to help the reader find the ranges of the targets to the arrays, but we agree that it may have been difficult to realize without appropriate guidance in the figure legend.

How did arrangement of the array change the sensitivity of the hydrophones?

We did not use the correct wording here. What we meant was the sensitivity at each point of the array, rather than hydrophone sensitivity. Sometimes hydrophones had to be replaced or switched around, especially when the array configuration was changed. We have clarified that in the revised version of the manuscript.

Is vertical axis tested?

Yes, the piston was fitted to all the hydrophones with peak levels exceeding the rms noise level of the channel by at least 14 dB (hydrophones marked with black filled circles in Figure 4–figure supplement 1).

*The subsection headed “Live prey capture”: Is this dataset (and*
Figure 2*) relevant for the study? Apparently, the data are not used for anything. The relevance would be obvious if both the array and the tag had recorded a trial in order to reveal possible correlation between changes in the echogram/acoustic scene and beam width, but that is not the case, as far as I can see*.

We do feel that it conveys a visual representation of the echoic complexity faced by echolocating animals and as such it is important in our argument that the adaptable FOV shown here may have evolved to deal with this complexity. We now refer to it (Discussion, first and third paragraphs) more often in the manuscript and believe it has illustrative value (i.e. that it helps visual creatures like ourselves better appreciate how a complex environment might be interpreted using echoes). Our intention was therefore to use this figure to demonstrate that when buzzing in a natural context porpoises can move through acoustic scenes of different complexities. While echograms have been published for beaked whales chasing prey, no such literature currently exists for harbour porpoises.

We do of course have echograms from the star-shaped array recordings (we used them for range estimations). They show that in the pen the array constitutes an acoustically cluttered environment. But the clutter did not vary between trials, since we used the same setup every time. Also, the animals catching fish in experiment one were not tagged, and we therefore cannot examine the clutter they experienced as a comparison for the star-shaped array data.

Do you really think Atlantic cod is a good TS for porpoise prey? Most would be much smaller and lower TS.

Cod is the second-most common prey species found in the stomachs of harbour porpoises by-caught in the Danish Straits (see Sveegaard 2011 for a review). It is true that adult cod reach sizes that cannot be consumed by harbour porpoises (according to Andreasen (2009) since 94% of porpoise prey is smaller than 45 cm). However, the Atlantic cod subjects used by [8] were 29-30 cm long (we have now added this information to the figure legend), and Börjesson et al. (2003) reported the average length of cod found in porpoise stomachs to be 28 cm, so we feel that this is a relevant comparison.

Figure 5
*is very interesting. The two trials are surprisingly different. Why is that? Were these recorded with the linear array or the star shaped? Probably star shaped, but what configuration*?

Both trials were recorded with the small star-shaped array. During the trial shown in the lower two panels, the porpoise was not blindfolded, which could have contributed to the differences between the trials. This information has been added to the figure legend.

Video 3*: The video shows depression in the whole buzz phase. If that is a proxy for broadening the view, it suggests a constant broad beam in the buzz and not variation as suggested by Results and Discussion in this manuscript*?

The video is of a porpoise stationed at a constant range to a stationary target. That could explain a relatively constant beam angle and may further explain why most studies so far have not reported much beam variation.

Please clarify what are the supplementary figures reporting and why.

Following your recommendations, we have removed one of the figure supplements (formerly Figure 3–figure supplement 3), we included one of the figure supplements in the manuscript as an independent figure (formerly figure 3–figure supplement 2, now Figure 4) and we have added a new figure supplement (Figure 6—figure supplement 1).

Consequently, we now have 2 figure supplements:

Figure 4—figure supplement 1: illustrates the different steps of the piston fitting procedure of experiment 2. We feel that it aids the understanding of the results and their limitations.

Figure 6—figure supplement 1: plots beamwidth against centroid frequency and the results of a physical model of frequency-dependence of beamwidth for static circular piston transducers of different sizes. The plot shows that the variation of centroid frequencies measured in the two experiments could not explain the observed beamwidth changes, and thereby supports our hypothesis that the beamwidth changes resulted from changes in the size of the effective radiating aperture.

*References*:

Linnenschmidt M., Wahlberg M., Hansen J.D. 2013 The modulation rate transfer function of a harbour porpoise (*Phocoena phocoena*). Journal of Comparative Physiology A 199(2), 115-126.

Wisniewska, D.M., Johnson, M., Nachtigall, P.E. and Madsen, P.T. (2014). Buzzing during biosonar-based interception of prey in the delphinids *Tursiops truncatus* and *Pseudorca crassidens*. The Journal of Experimental Biology 217, 4279-4282

Jensen F.H., Bejder L., Wahlberg M., Madsen P.T. 2009 Biosonar adjustments to target range of echolocating bottlenose dolphins (*Tursiops sp.)* in the wild. The Journal of Experimental Biology 212, 1078-1086.

Wahlberg M., Jensen F.H., Soto N.A., Beedholm K., Bejder L., Oliveira C., Rasmussen M., Simon M., Villadsgaard A., Madsen P.T. 2011 Source parameters of echolocation clicks from wild bottlenose dolphins (*Tursiops aduncus* and *Tursiops truncatus*). The Journal of the Acoustical Society of America 130(4), 2263-2274.

Jensen F.H., Rocco A., Mansur R.M., Smith B.D., Janik V.M., Madsen P.T. 2013 Clicking in shallow rivers: short-range echolocation of Irrawaddy and Ganges River dolphins in a shallow, acoustically complex habitat. PloS one 8(4), e59284. (doi:10.1371/journal.pone.0059284).

Sveegaard, S. 2011. Spatial and temporal distribution of harbour porpoises in relation to their prey. In Department of Arctic Environment, National Environmental Research Institute. Aarhus University, Denmark, 128.

Andreasen, H. 2009. Marsvinets (Phocoena phocoena) rolle som prædator i de danske farvande. University of Copenhagen, Denmark.

Börjesson, P., Berggren, P., and Ganning, B. 2003. Diet of harbour porpoises in the Kattegat and Skagerrak Seas: Accounting for individual variation and sample size. Marine Mammal Science 19:38-58.